

# An improved, time-efficient approach to extract accurate distance restraints for $NMR^2$ structure calculation.

Aditya Pokharna[a], Felix Torres[a], Harindranath Kadavath[a], Julien Orts*[b], and Roland Riek*[a]

[a]Laboratory of Physical Chemistry, ETH, Swiss Federal Institute of Technology, HCI F217, Vladimir-Prelog-Weg 2, 8093 Zürich, Switzerland

[b]University of Vienna, Faculty of Life Sciences, Department of Pharmaceutical Sciences, Althanstrasse 14, 2F 353, A-1090, Vienna, Austria

*Correspondence to : Roland Riek (roland.riek@phys.chem.ethz.ch) and Julien Orts (julien.orts@univie.ac.at)

**Abstract**

Exact Nuclear Overhauser Enhancement (eNOE) yields highly accurate, ensemble averaged $^1$H-$^1$H distance restraints with an accuracy of up to 0.1 Å for the multi-state structure determination of proteins as well as for Nuclear Magnetic Resonance Molecular Replacement ($NMR^2$) to determine the structure of the protein-ligand interaction site in a time-efficient manner. However, in the latter application, the acquired eNOEs lack the obtainable precision of 0.1 Å because of the asymmetrical nature of the filtered NOESY experiment used in $NMR^2$. This error is further propagated to the eNOE equations used to fit for and extract the distance restraints.

In this work, a new analysis method is proposed to obtain inter-molecular distance restraints from the filtered NOESY spectrum more accurately and intuitively by dividing the NOE cross-peak by the corresponding diagonal peak of the ligand. The method termed diagonal-normalized eNOEs was tested on the data acquired by Torres et al. (Torres et al., 2020) on the complex of PIN1 and a small, weak-binding phenylimidazole fragment. The diagonal-normalised eNOE derived distance restraints $NMR^2$ yielded the right orientation of the fragment in the binding pocket, and produced a structure that more closely resembles the benchmark X-ray structure (2XP6) (Potter et al., 2010) with an average heavy atom RMSD of 1.681 Å than the one produced with traditional $NMR^2$ with an average heavy atom RMSD of 3.628 Å, attributed to the higher precision of the evaluated distance restraints .





## 1 INTRODUCTION

Nuclear Magnetic Resonance Molecular Replacement (*NMR*[2]) is a hybrid approach to determine the structure
of protein-ligand complexes, utilising a previously determined structure (for example, a X-ray structure or
a structure from a protein homolog) of the target protein and combining it with the spatial information
extracted by solution state NMR to identify the binding pocket of the protein and the orientation of the ligand
inside it (Wälti and Orts, 2018). The major strength of the method is that one does not need to carry out
protein resonance assignment to obtain the complex structure. Using *NMR*[2], Orts et al. has been able to solve
the structure of various complexes (Torres et al., 2020); (Wälti and Orts, 2018); (Orts et al., 2016) accurately
(up to 1 *Å*) within a few days of measurement and analysis. The *NMR*[2] structure calculation workflow is
detailed in (Orts and Riek, 2020) and relies on acquiring precise inter-molecular distance restraints.
In *NMR*[2], the $^{13}$C, $^{15}$N-labelled protein and non-labelled ligand are mixed and measured together using
the F1-[$^{15}$N,$^{13}$C]-filtered [$^1$H,$^1$H]-NOESY experiment (Zwahlen et al., 1997) to extract the inter-molecular
NOE rates and the corresponding distances. This analysis is performed in an in-built module within CYANA
structure calculation software (Güntert and Buchner, 2015) called ENORA (Strotz et al., 2017). ENORA fits
the NOE buildup curves obtained at multiple mixing times to extract exact cross-relaxation rates (eNOEs)
which produces semi-accurate distance restraints with both upper and lower limit (Vögeli et al., 2009).
However, the precision of these inter-molecular distance restraints is much lower ( ∼20% higher tolerance
needed) (Strotz et al., 2015) than the bi-directional intra-molecular eNOEs, usually measured inside the
protein, from a series of $^{15}$N, $^{13}$C-resolved [$^1$H,$^1$H]-NOESY experiments that have a precision of 0.1 *Å*. The
lower precision is attributed to the imbalanced magnetisation pathway within the F1-$^{15}$N, $^{13}$C-filtered [$^1$H,$^1$H]-
NOESY experiment, the lack of a clean steady state magnetisation at the beginning of the experiment, the
unknown spin diffusion contribution (Kalk and Berendsen, 1976) and the complexity involved in extracting
distances within ENORA, which further propagates errors arising from the NOESY spectrum.
In this work, we present an alternative approach for extracting cross-relaxation rates from the filtered
2D NOESY spectra that forgoes the need for the sophisticated and time-intensive eNORA calculations and
produces more accurate distances. The complex used in this study is that of cis/trans isomerase PIN1
with a low molecular weight fragment, 4-Methyl-2-(3-methylphenyl)-1H-imidazole-5-carboxylic acid, whose
structure of the interaction site was solved by Torres et. al (Torres et al., 2020), in order to test the *NMR*[2]
method for weak binding small molecules. This fragment called Compound 1 in the paper by Torres et al.
(Torres et al., 2020) produces very few inter-molecular eNOEs to PIN1, due to its small size (comprising only
a few protons) and low binding affinity (260 *μ*M). This makes the de-novo determination of the right pose of
the ligand in the binding pocket using *NMR*[2] very challenging.
As we shall see, our approach has been successful in producing better restraints for the PIN1-Compound
1 complex than the standard procedure thereby predicting the right orientation of the ligand in the binding
pocket when compared with the X-ray structure of this complex (2XP6) (Potter et al., 2010), which serves as
a benchmark to ascertain the accuracy of the *NMR*[2] structures.

## 2 THEORY

Following the standard NMR theory of the NOESY experiment (Keepers and James, 1984), the proposed
analysis arises out of simple approximations made on the fundamental equations used to calculate eNOEs.
Every spin pair that produces a cross-peak can be assumed to form a two-spin system. The cross-relaxation
rate for a two-spin system (i and j) can be analytically given as (Vögeli, 2014); (Boelens et al., 1988):

$$\frac{I_{ij}(t)}{I_{ii}(0)} = \frac{I_{ji}(t)}{I_{jj}(0)} = \frac{-\sigma_{ij}}{\lambda_+ - \lambda_-}(\exp\{-\lambda_- t\} - \exp\{-\lambda_+ t\}) \tag{1}$$





⁵⁹ where $I_{ii}(t)$ and $I_{ij}(t)$ represent the peak intensity of the diagonal and the cross-peak in the NOESY
⁶⁰ spectrum respectively. The cross-relaxation rate, $\sigma_{ij}$, further depends on $\lambda_\pm$ which are a function of auto-
⁶¹ relaxation rates of the two spins, $\rho_i$ and $\rho_j$.

$$\lambda_\pm = \frac{\rho_i + \rho_j}{2} \pm \sqrt{\left(\frac{\rho_i - \rho_j}{2}\right)^2 + \sigma_{ij}^2} \tag{2}$$

The diagonal intensities can be approximated by a single-exponential decay, completely independent of
the auto- and cross-relaxation rates of the other spin:

$$I_{ii}(t) = I_{ii}(0)\exp\{-\rho_i t\} \tag{3}$$

⁶² Furthermore, under the assumption that $\rho_i \approx \rho_j = \rho$, which holds true for small- to medium-sized proteins,
⁶³ the exponential terms in Equation 1 can be expanded to the second-order as follows:

$$\exp\{-\lambda_\pm t\} = \exp\{-(\rho \pm \sigma)t\} = 1 - (\rho \pm \sigma)t + \frac{(\rho \pm \sigma)^2 t^2}{4}\ldots \tag{4}$$

Combining Equations 1, 3 and 4, the following expression can be obtained:

$$\boxed{\frac{I_{ij}(t)}{I_{ii}(t)} = -\sigma_{ij}t} \tag{5}$$

⁶⁴ This straightforward expression relates the cross-peak and diagonal intensities at mixing time, t, to the
⁶⁵ cross-relaxation rate. These quantities can be directly extracted from NOESY spectra recorded at multiple
⁶⁶ mixing times and fitted with a simple linear model to compute the cross-relaxation rate. This forgoes the
⁶⁷ need for invoking the ENORA module to fit the NOE build-ups. More importantly, it produces more accurate
⁶⁸ rates as it only involves directly fitting the experimentally-derived peak build-up intensities once. With the
⁶⁹ standard approach used in ENORA, the diagonal intensities are fitted in accordance with Equation 3 to
⁷⁰ extrapolate the auto-relaxation rate, $\rho_i$ and the initial magnetisation, $I_{ii}(0)$. The error introduced to these
⁷¹ quantities by imprecise fitting of Equation 3 and low SNR of diagonal peaks is propagated to Equations 1 and
⁷² 2, which are used to determine $\sigma_{ij}$. Furthermore, the imbalance inherent to the F1-[$^{15}$N,$^{13}$C]-filtered [$^{1}$H,$^{1}$H]-
⁷³ NOESY experiment and the missing $\rho_i$ contributes to the relative error. This error is also compounded in the
⁷⁴ eNORA approach as the peak intensity data is transformed and used in multiple fitting equations.
⁷⁵ The rates determined with the new method proposed here using Equation 5 is termed diagonal-
⁷⁶ normalised NOEs. However, there is a level of uncertainty still attached to the restraints extracted via
⁷⁷ this method because of the assumption, $\rho_i = \rho_j$, especially for large ligand-protein complexes with weak
⁷⁸ binding affinities, as $\rho_j$ might be an order of magnitude above $\rho_i$.
⁷⁹ A simple test was performed to quantify the uncertainty introduced by the above assumption to the
⁸⁰ extracted distances. It involved taking artificial distances (3 Å and 5 Å) between two spin pairs followed by
⁸¹ back calculating the value of the respective cross-relaxation rates. The obtained rates were fed to Equations
⁸² 1 and 2 with varying assumptions of the values of the auto-relaxation rates ($\rho_j$ and $\rho_i$). The ratios of
⁸³ magnetisation transfer $\frac{I_{ij}(t)}{I_{ii}(t)}$ were obtained at identical mixing times [40, 60, 90 and 120 ms], as used by Torres
⁸⁴ et al. (Torres et al., 2020) and fitted according to the Equation 5 in an attempt to reproduce the artificial
⁸⁵ distances.
⁸⁶ The results of the test are detailed in Figure A1 in the Appendix. At the ratio of $\frac{\rho_j}{\rho_i}$=10, the highest
⁸⁷ measured ratio, generally expected for the complex of a large protein and a small ligand, our method was
⁸⁸ able to reproduce the inter-molecular distance with an accuracy of 12.45% for both 3 Å and 5 Å. Hence, we
⁸⁹ propose a distance accuracy of ± ~10% for our approach. This distance accuracy lies between distances
⁹⁰ derived from bi-directional eNOEs (0%) and uni-directional eNOEs (20%) (Strotz et al., 2015).





## 3 RESULTS AND DISCUSSION

In order to evaluate the accuracy of the distance extraction method of diagonal-normalised NOEs the PIN1-Compound 1 complex introduced above is used. All the NMR experiments on the PIN1-Compound 1 complex were conducted and the subsequent resonance assignments were performed by Torres et al (Torres et al., 2020) (co-authors in this study). They resolved the structure of the binding pocket using $NMR^2$ on the inter-molecular, uni-directional, eNOE-derived distance restraints which have an expected accuracy of 20% (Strotz et al., 2015). In this work, we have used their data, recorded on $^{15}$N,$^{13}$C-filtered [$^1$H,$^1$H]-NOESY spectra, to evaluate the performance of the diagonal-normalised eNOE analysis as compared to the the standard eNOE approach (Vögeli et al., 2009); (Vögeli, 2014).

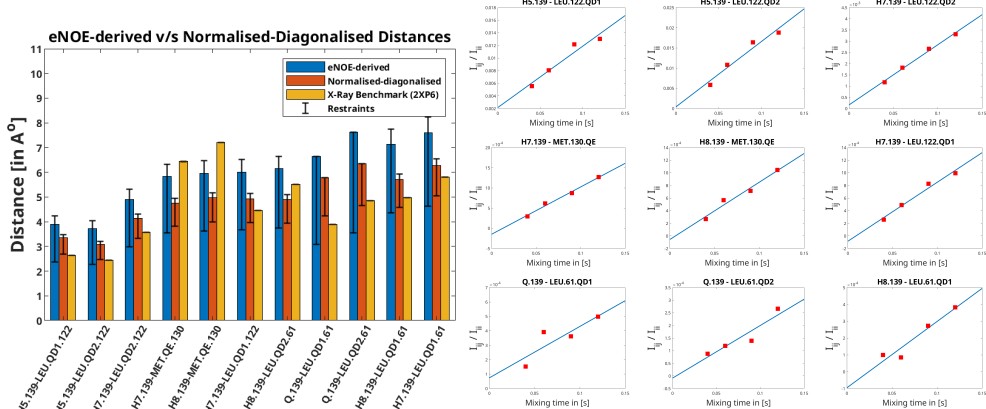

Figure 1: (Left) Distances extracted from F1-[$^{15}$N,$^{13}$C]-filtered [$^1$H,$^1$H]-NOESY using the eNOE (blue) and diagonal-normalised approach (red) compared to the benchmark X-ray structure. The bars denote the distances that arise from the cross-relaxation rates from the complex of PIN1 with Compound 1, as given in (Torres et al., 2020). The bars in yellow represent the distances back-calculated from the X-ray structure (2XP6) (Potter et al., 2010). The error bars denote the upper and lower limit restraints produced in CYANA (Güntert and Buchner, 2015) for the extracted distances. A tolerance of 20% and 10% was taken and for the eNOE and the diagonal-normalised approach extracted distances respectively. (Right) The ratio of NOE buildups to the corresponding diagonal peak intensities plotted against mixing time for the PIN1-Compound 1 complex for the diagonal-normalised approach. The data points were fitted using a linear least square fitting model in MATLAB (MATLAB, 2018). The slope denotes the cross-relaxation rate of the given peak, as per Equation 5.

Apart from being more time-efficient and intuitive, this method should also provide more accurate distances, as discussed in the Theory section. The NOE build ups plots, fit linearly according to Equation 5, are depicted in Figure 1 (right). The linear fits mostly tend to zero when mixing time is zero and the experimental data fits well even at longer mixing times for all cross-peaks. This indicates a lack of significant spin diffusion contribution. Moreover, it is easier to detect spin diffusion with this method compared to the standard approach using eNORA, as it manifests itself as non-linearity in the fitted data. This difference is illustrated in Figure A2 in the Appendix.

The derived distance restraints are also plotted against the conventional eNOE-derived distance restraint and the distances back-calculated from the benchmark X-ray structure (2XP6) (Potter et al., 2010) in Figure 1 (left) (The protons were added to the X-ray structure in CYANA (Güntert and Buchner, 2015)). Indeed, the diagonal-normalised distance restraints better resemble the ones from the X-ray structure (mean difference in the distances being 1.04 ± 0.65 $Å$) than the ones from the standard approach (mean difference in the distances being 1.57 ± 0.73 $Å$). The only exceptions being the distances that include the protons from the





solvent-exposed Methionine 130. The floppy nature of this region of the binding pocket is predicted to give
minimal distance data.
The inter-molecular distances obtained from the PIN1-Compound 1 complex through the conventional
eNORA-based method and the diagonal-normalised approach are plotted in Figure 1. The plots illustrate
that the restraints obtained via the latter are tighter by 0.4-1.2 Å. The source of this difference, as discussed
in the Theory section, arises from the inherent complexity involved in extracting distances from a filtered
2D-NOESY spectrum.
To evaluate the 10% error estimate deduced in the Theory section further and to study the impact of the
diagonal-normalised distance restraints on $NMR^2$ structure determination, $NMR^2$ structures of the complex
PIN1-Compound 1 were calculated with varying degree of precision of the diagonal-normalised distance
restraints (i.e. 0%, 5%, 10%, and 20%) (Table 1). The restraints were input in the $NMR^2$ algorithm and the
output structures were compared to the structure determined in (Torres et al., 2020) using standard eNOEs.
The $NMR^2$ program screens all potential combinations of methyl groups in protein and protons on the
ligand and calculates the complex structure for all of the possibilities without needing protein assignment.
The success of an $NMR^2$ run lies in it being able to discriminate between all the possible structures and
pinpoint the right pose of ligand in the binding pocket. This is especially difficult for small fragments like
Compound 1, with only 5 distinct protons/methyl groups.

Table 1: Table detailing the results of $NMR^2$ calculations with distance restraints extracted from eNOE and diagonal-normalised method with varying values of errUni in CYANA.

| Method Used | Precision (in % of the given distance)[1] | Does the structure converge up to TF = 2 Å² ? (Yes/No)[2] | Target Function of 4 lowest energy conformers (in Å²) | Total number of degenerate lowest energy conformers | RMSD w.r.t to the benchmark (2XP6) (in Å²) |
|---|---|---|---|---|---|
| eNORA-based | 20% | Yes | [ 0, 0, 0, 0 ] | 10+ | 3.63 |
| Diagonal-Normalised | 20% | Yes | [ 0, 0, 0, 0 ] | 5 | 2.17 |
| Diagonal-Normalised | 10% | Yes | [ 0.03,0.12,0.20,0.73 ] | 1 | 1.68 |
| Diagonal-Normalised | ≤ 5% | No | – | – | – |

[1]A precision of x% dictates the value of upper limit and lower limit distance restraints with the upper limit distance restraint being (1+ x%)*(extracted distance) and the lower limit distance restraint being (1- x%)*(extracted distance).

[2] A target function of less than 2 Å² within $NMR^2$ is considered a successful structure determination (Orts and Riek, 2020).





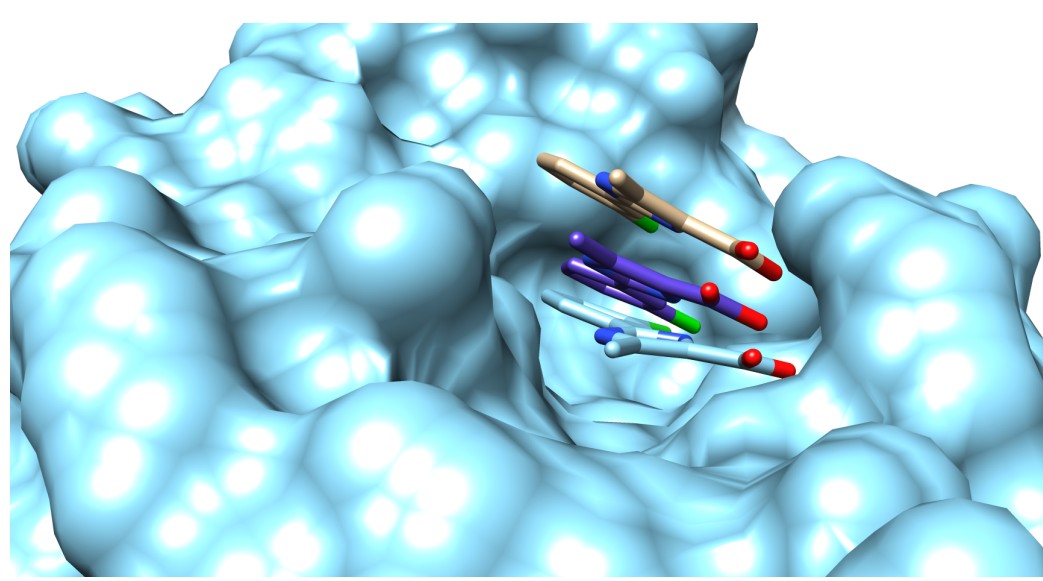

Figure 2: Surface representation of the binding pocket of PIN1 with Compound 1. Coloured in cyan is the surface of the structure determined by X-ray crystallography studies (2XP6) (Potter et al., 2010). Coloured in brown is the structure determined by Torres et al.(Torres et al., 2020) with a distance precision of 20% using the standard ENORA approach and coloured in purple is the structure determined by $NMR^2$ calculations using the distances extracted via the diagonal-normalised approach with a precision of 10%. The nitrogen, oxygen and chlorine atoms on the ligand are coloured blue, red and green respectively.

Table 1 outlines the details of the structure calculation test. The restraints obtained through the eNORA-based method were not good enough and gave rise to more than 10 degenerate structures with a Target Function (TF) of 0 $Å^2$, meaning that all experimental distance restraints were fulfilled without inconsistency/error in any of the 10 degenerate structures. The structure in which the ligand has the same orientation inside the binding pocket, as the benchmark X-ray structure (2XP6) (Potter et al., 2010), has an RMSD of 3.63 $Å$ with respect to the X-ray structure (2XP6). Using the diagonal-normalised distance determination procedure with a precision of 20%, a better performance is observed with only 5 degenerate structures with a TF of 0 $Å^2$, which included the complex structure with Compound 1 in the right pose (RMSD of 2.17$Å$). For the anticipated precision of the distance restraints of 10%, the calculation produced only one structure with a TF = 0.03 $Å^2$, which shows the same orientation as the crystal structure with an RMSD of 1.68 $Å$. This structure superimposes well with the benchmark structure, as shown in Figure 2. A visual inspection of the binding pocket illustrated in Figure 2 shows that the ligand appears deeper in the binding pocket and better aligned with the crystal structure compared to the structure obtained by traditional, eNORA-based $NMR^2$. For a distance precision of 5% and below, the calculations did not converge to structures that fulfil the experimental restraints and produce structures below the hard limit of TF < 20 $Å^2$. This is expected since the distance restraints are not of the quality of bidirectional restraints due to the assumption $\rho_i = \rho_j$, the lack of spin diffusion correction and other restrictions inherent to the $NMR^2$ protocol, such as the use of a previously determined protein structure and combining X-ray and NMR data.

The strength of this approach lies in distinguishing the correct pose of a weak-binding, low molecular weight ligands which gives very few inter-molecular NOEs inside the binding pocket of a larger proteins. Nevertheless, this approach was also tested on the protein-ligand complex of HDM2, a human oncogenic protein, with caylin-1, which presents abundant inter-molecular NOEs. The traditional eNORA-based $NMR^2$ was successful in characterising the structure of protein-ligand interaction site (7QDQ), as shown in the

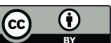



work of Mertens et al. (Mertens et al., 2022). With othe diagonal-normalised approach at 10% precision,
we obtained the same pose of caylin-1 in the HDM2 binding site, as Mertens et al., with a TF of 1.52 $Å^2$
and RMSD between the two structures being 0.81 $Å$ (refer to Figure A3 in the Appendix). Furthermore,
the calculations made with 15% and 20% precision also matched the predictions of traditional $NMR^2$ in
identifying the right structure. This is further evidence that our approach can at least match the predictions
of traditional $NMR^2$ in the case of strong binders and possibly exceed them in the case of weak binders with
less NOEs.
To sum up, this work proposes an intuitive and time-efficient, alternative method to extract precise
distance restraints from a series of filtered-NOESY spectra, that gives, in the system studied, an accurate
$NMR^2$ structure of the protein-ligand interaction site.

## 4  MATERIALS AND METHOD

No new material was prepared for the sake of this work. The protocol to express and purify the protein and
the ligand and to mix them afterwards is detailed in (Torres et al., 2020).
No new NMR experiments were conducted for this work either. The peak intensities from the spectra
acquired by Torres et al. were extracted via ccpNMR (Skinner et al., 2016). The intensities were later fit to
acquire the rates in the MATLAB Software suite (MATLAB, 2018). The structure calculation were performed
by $NMR^2$ program through CYANA (Güntert and Buchner, 2015). All the structures were displayed and
overlaid over each other using the Chimera molecular visualisation tool (Pettersen et al., 2004).

## 5  ACKNOWLEDGEMENT

We would like to thank the Swiss National Foundation for financial support through the grant number
310030_192646.

## 6  CONFLICT OF INTEREST

No conflict of interest.

## 7  DATA AVAILABILITY

All relevant data can be obtained upon request.

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



8   APPENDIX

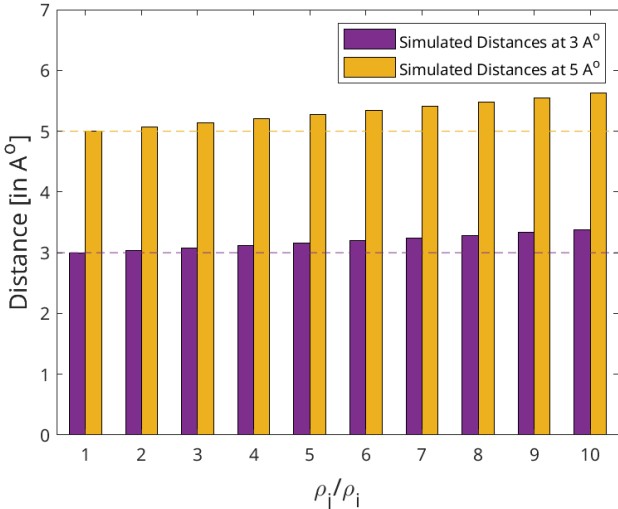

Figure A1: The effect of the relative auto-relaxation rates of the protein and the ligand on the distances extracted via the diagonal-normalised approach. The dotted lines represent the artificial distances $3\text{Å}$ and $5\text{Å}$ in purple and yellow respectively. The corresponding bars denote the distances back-calculated using the diagonal-normalised approach from the artificial distances depending on the the relative auto-relaxation rates, $\rho_i$ and $\rho_j$. Each set of distances (bars) are derived through varying assumptions of the values of $\rho_i$ and $\rho_j$ with respect to each other ranging from $\frac{\rho_j}{\rho_i} = [1 \text{ to } 10]$.

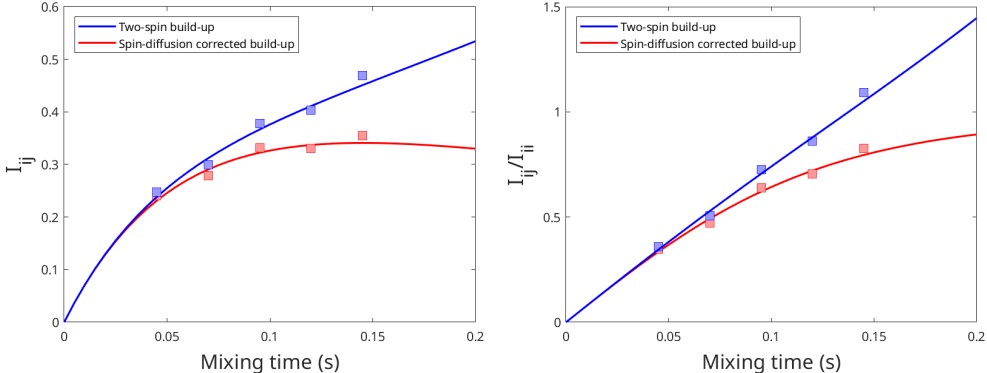

Figure A2: Effect of spin diffusion on the intensity build-up curves produced from eNORA-based approach (left) and diagonal-normalised approach (right). The build-up curves were fitted using artificially simulated peak intensities in a model system. The blue curve represents the intensity build-up in an isolated two-spin system and the red curve represents the same two spins experiencing spin-diffusion due to presence of other spins in the system. The comparison between the plots highlight that it is easier to detect the influence of spin-diffusion with the diagonal-normalised approach (right), as it induces deviation from the expected linear fit.

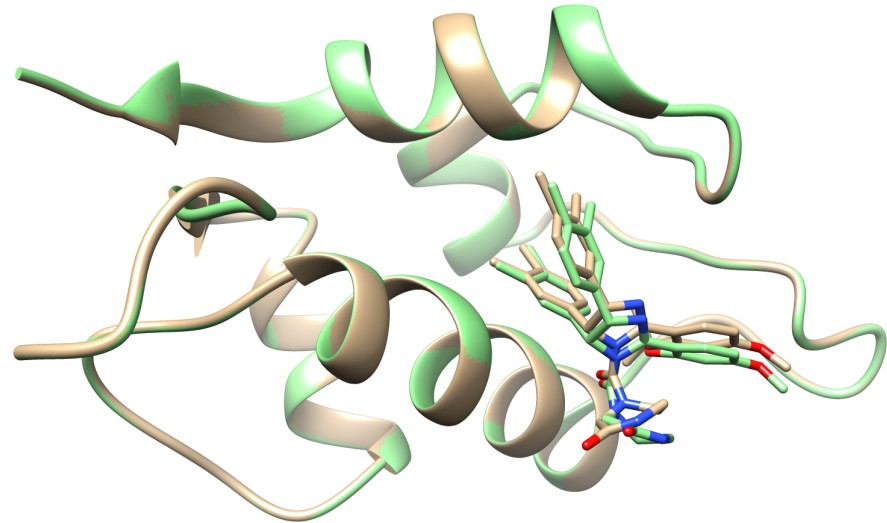

Figure A3: Ribbon representation of the protein, HDM2, with the stick representation of caylin-1 present in the binding pocket. Coloured in brown is the surface of the structure determined by traditional $NMR^2$ (7QDQ) by Mertens et al. (Mertens et al., 2022). Coloured in green is the structure determined by $NMR^2$ calculations using the the diagonal-normalised approach with a precision of 10%. The nitrogen and oxygen atoms on the ligand are coloured blue and red respectively.