# Peer review of "An improved, time-efficient approach to extract accurate distance restraints for *N*MR2 structure calculation."

_Magnetic Resonance, 2022_

## Author Comment (AC1)

**Reviewer 1:**

*We would like to thank the reviewer 1 for his suggestions to improve the manuscript. Answers to the questions of reviewer 1are stated after each request.*

Reviewer 1: The manuscript entitled "An improved, time-efficient approach to extract accurate distance restraints for NMR2 structure calculation" by

Pokharna, Torres, Kadavath, Orts, and Riek describes a new approach for improving the precision of distances measured by NMR and used to calculate the position of ligand in protein pocket. The proposed method which consist to normalize the measured NOEs with the intensities of the corresponding diagonal element, has been proposed in the past, but is validated here in the frame of the NMR2 approach, used to determine poses of ligands in a protein pocket using NMR measurements. The normalisation with the diagonal intensity is shown to improve the precision of the determination of a ligand pose.

The content of the manuscript is significant and deserves publication. Nevertheless, at several places, the manuscript text should be worked
out to clarify the presentation of the work:

a) the last abstract sentence is very complicated and fuzzy.

The diagonal-normalised eNOE derived distance restraints NMR2 yielded the right orientation of the fragment in the binding pocket, and produced a structure that more closely resembles the benchmark X-ray structure (2XP6) (Potter et al., 2010) with an average heavy atom RMSD of 1.681 Å than the one produced with traditional NMR2 with an average heavy atom RMSD of 3.628 Å, attributed to the higher precision of the evaluated distance restraints.

*Answer: We followed the suggestion of the reviewer. The last sentence in the abstract reads now: "NMR2 calculations performed using the distances derived from diagonal-normalised eNOEs yielded the right orientation of the fragment in the binding pocket, and produced a structure that more closely resembles the benchmark X-ray structure (2XP6) (Potter et al., 2010) with an average heavy atom RMSD of 1.681 Å with respect to it, when compared to the one produced with traditional NMR2 with an average heavy atom RMSD of 3.628 Å. This is attributed to the higher precision of the evaluated distance restraints."*

Reviewer 1: b) line 90: the expression "(0%)" is a bit strange, as no experimental measure can be obtained at a infinite precision.

*Answer: We agree with the reviewer that no experimental measure can be obtained at infinite precision. The 0% must be read in the context of the implementation within the software CYANA which uses a harmonic potential as penalty function (i.e. target function TF) taken into account the lack of precision. In the revised version of the manuscript the following sentence is added around line 90: "It is noted that the software CYANA uses a harmonic potential for a its target function (TF) to accommodate the distance restraints and as such 0% means the harmonic potential only, while 20% distance tolerance means a flat potential from 0-20% distance followed by the harmonic potential beyond.*

Reviewer 1: c) in Figure A2, results of simulations are described. It is not clear on which system the simulations were performed.

*Answer: We adjusted the figure caption to "Figure A3: Effect of spin diffusion on the intensity build-up curves produced from eNORA-based approach (left) and diagonal-normalised approach (right). The build-up curves were fitted using artificially simulated peak intensities in a model system, Third Immunoglobulin Binding Domain of Protein G (GB3), which has been extensively studied using eNOE spectroscopy. The blue curve represents the intensity build-up in an isolated two-spin system with an inter-proton distance of 3.83 Å and the red curve represents the same two spins experiencing spin-diffusion due to presence of other spins in the system. The comparison between the plots highlight that it is easier to detect the influence of spin-diffusion with the diagonal-normalised approach (right), as it induces deviation from the expected linear fit."*

Reviewer 1:

d) caption of Figure 2: it is not clear that the colours purple and brown concern the ligand. The correspondence between the ligands shown in

Figure 2 and the lines of Table 1 should be given.

In addition, to which pose corresponds the ligands in which all atoms are coloured? (I don't understand this part of the question.)

In lines 130-147, the description of Figure 2 is not clear, the ligand colors should be quoted.

*Answer: Following the suggestions of the reviewer, we edited the manuscript as requested.*

Reviewer 1: e) What is the meaning of Total number of degenerate lowest energy conformers in Table 1?

*Answer: This is the total no. of distinct orientations of the ligand within the binding pocket that give a CYANA TF of 0 Å after NMR2 calculations. A number higher than 1 means that the experimental restraints were not sufficient in quality for the method to discern between these orientations. In footnote 3, the following sentence is added in the revised version of the manuscript: "The Total number of degenerate lowest energy conformers is the total number of distinct orientations of the ligand within the binding pocket that was obtained with a CYANA TF of 0 Å from NMR2 calculations."*

Reviewer 1: In addition, there is a more basic question about the method NMR2. The manuscript presents the use of NMR2 for the determination of very
precise position of ligand, which should correspond to an high affinity interaction? But, NMR is used for studying the interaction of low
affinity ligands for which the poses may much less precise. It would be interesting to insert comments about these points.

*Answer: This is an interesting point raised by the reviewer that we would like to answer from two points of view. We demonstrated here and by Torres et al. (2020) that the NMR2 approach presented is able to identify the binding pocket also of weak binders. On the other hand, there might be indeed as suggested multiple binding configurations and conformations present which we study currently with eNOE-based multi-state structure determination that requires the entire assignment and both inter- and intra-molecular eNOE-based distance restraints. Currently the NMR2 approach does not include multi-state structure determination at standard NMR resolution. We may implement that feature in the future, but that would most likely require even better input data.*

*For the determination of multiple binding configurations, namely multi-state structure determination, at high resolution, a detailed eNOE-based analysis would be required. In the revised version of the manuscript at the end of line 149, the following sentence is stated. It is noted that the presence of multiple configurations/conformations of the ligand in the binding pose will require detailed eNOE-based multi-state structure calculations (Vögeli et al. 2012, Ashkinadze et al. 2021)*

---

## Author Comment (AC2)

**Reviewer 2:**

*We would like to thank the reviewer 2 for his suggestions to improve the manuscript. Answers to the questions of reviewer 2 are stated after each request.*

This manuscript describes an extension of recent work by the same group on the method they call NMR molecular replacement, or NMR2. As they have described in previous papers cited in this manuscript, in the NMR2 approach the structure of a protein-ligand complex in solution is determined using a combination of intermolecular NOE data from the complex and a previously determined structure (often a crystal structure) of the free protein. Much of the interest in this method derives from its potential for use in drug discovery; particularly since the method does not involve making independent assignments for the protein NMR resonances it can in principle provide information on a timescale that is useful in the context of a ligand screening program.

However, in the context of a fragment-based drug development strategy the ligand fragments tested will often be very small molecules, with weak binding affinities and few NMR resonances, implying that the NOE data are very sparse and possibly difficult to measure. A related issue is that distance calibration of a limited set of intermolecular NOEs poses significant challenges, as there are no fixed, known intermolecular distances that can be used as standards (as would be done for intramolecular NOE calibration) and the methods employed instead, such as the ENORA approach previously published by members of the same group, are less direct and may give inaccurate distance restraints. In this manuscript the authors describe an alternative approach for calibrating intermolecular distance restraints, based on comparison of cross-peak intensities with those of diagonal peaks in the same spectrum. This results in shorter distance restraints than those obtained using ENORA, which, at least in the test system used in this manuscript, results in the ligand being pulled more deeply into the binding pocket on the protein, although still not as deeply as is seen in the related crystal structure of a closely-related complex in this case. This may mean the new solution structure resembles the true structure more closely than did the previous solution structure, though one should probably not assume that the true solution structure is identical to the crystal structure for such a weak complex.

I think the manuscript potentially represents a worthwhile step forward and should be published, subject to the following points being satisfactorily addressed:

Reviewer 2: 1) The manuscript describes results for one ligand, called Compound 1, but there seems to be some confusion in Figure 2 since this shows a different ligand, at least in the case of the crystal structure. The earlier paper from the same group by Torres et al. (2020) employing the ENORA approach describes results for three closely related ligands that differ only in the substituent at position 3 of the benzene ring: in Compound 1 it is CH3, in Compound 2 it is Cl and in Compound 3 it is CF3. The crystal structure PDB 2XP6 on which Figure 2 is based definitely contains Compound 2, yet the caption to Figure 2 in the present manuscript refers only to Compound 1. Moreover, all three copies of the ligand shown in the figure have a green atom at position 3 of the benzene ring that the caption states represents chlorine, but this cannot be correct for Compound 1, which was the subject of all the NMR calculations in the present manuscript. These discrepancies need to be resolved. Maybe this is just a colouring mistake when preparing the figure, but even if this is corrected the wording in main text and caption needs to be adjusted to clarify that the crystal structure contains Compound 2, not 1, and consequently the comparison shown is not quite as direct as it currently seems. For instance, the affinity of Compound 2 is weaker than that of Compound 1, which could perhaps influence how deeply into the binding site it binds.

*Answer: We are very sorry for the mistake on our side and would like to express our thank for the careful reading of the reviewer since it is indeed Compound 2 not 1 in the Torres et al. manuscript. In the revised version of the manuscript, the IUPAC name and binding affinity has been changed to reflect that.*

2) Torres et al. (2020) states that for NMR data collection for the complex of PIN1 with Compound 1, the protein concentration was 1.3 mM, the ligand concentration was 2.5 mM and the Kd was 260 uM. Using the quadratic binding equation for an A + B <-> AB equilibrium, it is straightforward to calculate that under these conditions the protein is 84% in the bound state and the ligand is 44% in the bound state. Figures S2 and S3 of Torres et al. make clear that, as would be expected, the system is in fast exchange on the chemical shift timescale, so the NMR signals used in the NOE experiments are all averages that contain substantial contributions from both the free and bound states, in line with the populations given above. However, eqs 1 to 5 make no reference to this, and as far as I can see from the cited papers (mainly Vögeli 2014) were originally derived for intramolecular systems where the complication of a free-bound

equilibrium does not exist.  Is it justified to use those equations without modification when dealing with an exchanging system where a substantial proportion of the observed signals is contributed by the free states?  Even if it is, I think this point needs to be carefully explained (e.g. what assumptions/definitions does it rely on?) and justified with appropriate citations in the paper.  (Also, though this is not an issue with the present manuscript, it is not fully clear to me either how the presence of exchanging bound and free states was accounted for in the ENORA approach in the Torres et al. paper - unless I have misunderstood it was done by calculating an "effective correlation time" for the complex that is a population-weighted average, but that does not quite make physical sense to me, nor is the calculation fully described in that paper).  The step of converting the cross-relaxation rates into distance restraints is also not described in the present paper; I feel that at the least there should be a citation given that makes clear how this was done.

*In NMR2, the 13C, 15N-labelled protein and non-labelled ligand are mixed and measured together using the F1-[15N,13C]-filtered [1H,1H]-NOESY experiment to extract the inter-molecular NOE rates. The inter-molecular NOEs should account for contribution from the free state due to the 'effective' correlation time used.*

*The theory for effective correlation time calculation comes from the review (Recent developments in transferred NOE methods, Ni F., 1994)).*

*The conversion from cross-relaxation rates to distances can be made via the equations reported in the previous NMR2 publications (see below). Please also note that the effective correlation time in the case of Pin1 was derived from steric distances found in the fragments. (This has been added to the revised manuscript)*

*The NMR2 Method to Determine Rapidly the*
*Structure of the Binding Pocket of a Protein–Ligand*
*Complex with High Accuracy, Magnetochemistry 2018, 4, 12; doi:10.3390/magnetochemistry4010012*

Reviewer 2: 3)  Although the authors did also test their new method with a second example (binding of caylin-1 to HDM2), as they point out themselves that was a less demanding case where the new approach made little difference to the outcome.  However, in the Torres et al. 2020 paper the authors applied the original ENORA approach to the PIN1 complexes of the two more weakly binding ligands, Compounds 2 and 3, in addition to that of Compound 1.  In those cases they found the ENORA method did not converge unless they assumed the assignments of the protein methyl signals receiving intermolecular NOEs were the same as those for the (better behaved) complex of Compound 1.  An obvious question would therefore be, does the new method based on diagonal-normalised restraints improve the situation sufficiently that the complexes of Compounds 2 and 3 can be solved without having to use the methyl assignments from the Compound 1 complex?  Did the authors try this, and if so what did they find?

*Answer: The method was used on complexes involving Compound 2 and 3 from the Torres et al. (2020) as neither of them converged upon a unique structure with traditional NMR2. The results from PIN1-Compound 2 are presented in our work and unlike traditional NMR2, diagonal-normalised method at 10% precision was able to pinpoint the right orientation of the ligand. As mentioned earlier, we had mistakenly labelled Compound 2 as Compound 1.*

*As for PIN1-Compound 3 complex, it failed to converge on a unique structure, possibly due to insufficient number of distance restraints.  However, we noticed that the diagonal-normalised approach predicted fewer structures with TF=0 as compared to traditional NMR2. Furthermore, there was no X-ray structure available of the PIN1-Compound 3 complex for direct comparison and thus we did not report on it in the manuscript.*

Reviewer 2: 4)  Looking at Figure 1, it is striking that the intermolecular NOE restraints involving the methyl of methionine 130 are significantly shorter than the corresponding distances in the crystal structure, whereas all the other such restraints are longer than the corresponding crystal distances.  The authors mention this briefly, noting that the methionine is in a solution-exposed "floppy" region, but I did not find this a very plausible explanation; in my own experience floppy regions generally give weaker NOEs, not stronger ones.  Is there perhaps a real difference between the crystal and solution structures?  For instance, might it be that the methionine methyl has a hydrophobic interaction involving the methyl group of Compound 1 (the ligand used in the NMR experiments) that is missing in the complex of Compound 2 (the ligand present in the crystal structure) where the corresponding substituent is a chlorine?  Looking at PDB 2XP6, it seems entirely plausible that the methionine sidechain could move to a position closer to the methyl group and that this could indeed shorten the two distances involved in the restraints in question.  This could also perhaps be at least part of the reason why Compound 1 binds more tightly than Compound 2.

*Answer: The point raised by the reviewer is interesting while it can not be answered without orthogonal data acquisition out of the scope of the present technical note. In an earlier work we showed for a different system that the NMR2 structure was correct while the x-ray complex structure was incorrect attributed to crystal packing artifacts. The orthogonal approach was finding an alternative crystal packing artifact-free crystal. We also agree with the reviewer that the statement "The floppy nature of this region of the binding pocket is predicted to give minimal distance data" is not a satisfactory explanation and thus we removed the text in the revised version of the manuscript.*

Reviewer 2: 5)  It would be useful to show a line drawing of the structure of Compound 1, and perhaps also of Compound 2, to help clarify the explanation of the system (not all readers will be able immediately to convert the IUPAC name given in the introduction to a structure).  Also, it would be very useful to show a second orientation of the structural representation in Figure 2, as in the current view it is almost impossible to see how far into the binding pocket the two sets of NMR-based calculations have actually pulled the ligand.  A rotation of +60° about x would probably achieve a clearer view of this.

*Answer: Following the suggestions of the reviewer, a ChemSpider-based figure of the ligand is shown in Figure A1. In addition an alternate orientation is also shown in Figure 2.*

Reviewer 2: 6)  While using diagonal peak intensities may bring advantages in the analysis, it requires that the diagonal peaks are separately resolved, thus loosing the resolution advantage of separating the peaks in multiple dimensions.  Is this likely to be a difficulty in more general applications of the new approach, or does the expected sparsity of ligand signals mean there is unlikely to be a problem?

*Answer: As the reviewer mentioned himself, the method is indeed used and useful for small ligands that are interesting for drug research and do not usually show severe chemical shift overlap.*